# Prolonged ovarian storage of mature *Drosophila* oocytes dramatically increases meiotic spindle instability

**Ethan J Greenblatt, Rebecca Obniski, Claire Mical, Allan C Spradling\***

Department of Embryology, Howard Hughes Medical Institute Research Laboratories, Carnegie Institution for Science, Baltimore, United States

**Abstract** Human oocytes frequently generate aneuploid embryos that subsequently miscarry. In contrast, *Drosophila* oocytes from outbred laboratory stocks develop fully regardless of maternal age. Since mature *Drosophila* oocytes are not extensively stored in the ovary under laboratory conditions like they are in the wild, we developed a system to investigate how storage affects oocyte quality. The developmental capacity of stored mature Drosophila oocytes decays in a precise manner over 14 days at 25°C. These oocytes are transcriptionally inactive and persist using ongoing translation of stored mRNAs. Ribosome profiling revealed a progressive 2.3-fold decline in average translational efficiency during storage that correlates with oocyte functional decay. Although normal bipolar meiotic spindles predominate during the first week, oocytes stored for longer periods increasingly show tripolar, monopolar and other spindle defects, and give rise to embryos that fail to develop due to aneuploidy. Thus, meiotic chromosome segregation in mature *Drosophila* oocytes is uniquely sensitive to prolonged storage. Our work suggests the chromosome instability of human embryos could be mitigated by reducing the period of time mature human oocytes are stored in the ovary prior to ovulation.

**\*For correspondence:**
spradling@ciwemb.edu

## Introduction

Animal oocytes grow extensively to become the largest body cells, but at a few specific stages ovarian follicles can persist in a non-growing state. Following recombination and prophase arrest at the diplotene stage of meiosis, mammalian oocytes within primordial follicles cease demonstrable development to establish the 'ovarian reserve,' whose slow utilization over multiple decades in humans determines the duration of female fertility. Both oocytes and granulosa cells within primordial follicles remain able to transcribe and translate genes and are bathed in maternal nutrients, which may assist in maintaining their long period of quiescence. Eventually, arrested oocytes resume growth and develop to their final size while remaining in meiotic diplotene. Shortly before fertilization, oocytes mature, during which meiosis resumes and progresses to an arrest at metaphase I or II (*Coticchio et al., 2015*; *Hughes et al., 2018*).

In many species, fully grown oocytes also have a period of quiescence. In a good nutritional environment, which is common in the laboratory but rare and transient in the wild, mated *Drosophila* females ovulate mature oocytes shortly after they reach their final size. However, *Drosophila* store metaphase I-arrested oocytes for multiple days if adequate protein or sperm are unavailable, despite a lack of transcription. Analysis of polysomes suggests that stored oocytes maintain protein production, though at a reduced level (*Lovett and Goldstein, 1977*). Likewise, mammalian oocytes routinely cease transcription and become quiescent sometime after reaching their full size (*Abe et al., 2010*; *Jukam et al., 2017*). Oocytes remain transcriptionally inactive until zygotic genome activation at the two-cell stage (mouse) or at the 4-cell stage (human). It has been difficult to study the exact

duration and biological significance of mature oocyte storage in mammals because of asynchrony and oocyte to oocyte variation (reviewed in *Conti and Franciosi, 2018*).

Storing oocytes is generally associated with a significant risk of functional impairment. In humans, where all oocytes are stored to some extent, a portion of oocytes develop meiotic segregation errors including non-disjunction that are the major cause of miscarriage. Past the age of 35, chromosome mis-segregation further increases as reflected in exponentially growing rates of Down's syndrome (*Webster and Schuh, 2017*). However, studies of in vitro fertilized human oocytes suggest that spindle-related errors in mitotic chromosome segregation during early embryonic cell cycles are frequent even in embryos derived from donor eggs of young women (*McCoy et al., 2015*). The high frequency of meiotic defects in human oocytes has been explained by the exceptional length of time they spend as arrested primordial follicles after the establishment of sister chromatid cohesion (*Chiang et al., 2010*; *Herbert et al., 2015*). In *Drosophila* oocytes, genetic studies also support a role of cohesion loss in meiotic chromosome instability (*Hughes et al., 2018*; *Subramanian and Bickel, 2008*). However, cohesion loss may not fully explain the high frequency of non-disjunction, and evidence in mice supports the proposal that altered microtubule dynamics leading to aberrant spindle formation also contributes to non-disjunction (*Nakagawa and FitzHarris, 2017*).

Here, we show that mature *Drosophila* oocytes remain capable of supporting embryonic development for many days while stored in the ovary, providing a system for the molecular genetic analysis of oocyte aging. Oocytes stored only briefly develop with high fidelity. However, as aging continues, completing meiosis successfully following fertilization becomes the major factor limiting oocyte viability. Cytologically detectable spindle defects increase during storage and early developmental arrest gradually become the predominant fate of the resulting embryos. Translation of mRNAs encoding meiotic metaphase and spindle-related proteins decline as part of a general 2.3-fold reduction during aging in the absence of bulk changes to mRNA levels. Our findings show that storage of highly functional mature oocytes in vivo is sufficient to destabilize chromosome segregation, suggesting that the prolonged storage of mature oocytes may be an important source of meiotic chromosome instability in human females.

## Results

### A general method for studying *Drosophila* oocyte aging

*Drosophila* ovaries are organized into highly regulated ovarioles that preserve the order in which follicles develop (*Figure 1A*). Ovarian biology allowed us to develop a method to obtain mature oocytes that have been stored in the ovary for a known period of time. Newly eclosed virgin female flies with immature ovaries are fed a nutrient-rich yeast paste that stimulates exactly two young follicles per ovariole to develop to maturity past a nutrient-sensitive checkpoint at stage 8 (*Figure 1A, B*). Withdrawal of the yeast food after 24 hr prevents any additional follicles from passing the checkpoint, however oocyte and maternal physiology are not adversely affected (*Drummond-Barbosa and Spradling, 2001*). In the absence of mating, the mature eggs are stored in the ovary indefinitely and not replaced, as shown by the continuing absence of post-checkpoint stage 10 oocytes (*Figure 1B*; *Greenblatt and Spradling, 2018*).

We measured the stability of stored oocytes over time by placing females with held, mature oocytes of known age with males, which stimulates the rapid fertilization and deposition of their mature oocytes following mating (*Figure 1C*). We found that while oocytes stored for 5 days or less support development to hatching at high rates (90–96%), more extensively stored oocytes generate embryos with lower levels of hatching. Loss of developmental capacity follows highly reproducible sigmoidal kinetics over the course of 1–4 weeks depending on temperature (*Figure 1D*). At 29°C, 25°C or 20°C, 50% of eggs fail to hatch after about 7, 12, or 23 days of storage, respectively (*Figure 1D*). Thus, by appropriately feeding newly eclosed females and delaying mating, we are able to obtain an abundant supply of oocytes for study of known age that are at a known point on an aging curve. Stored mature oocytes are thought to be impermeable to macromolecules; they showed no visible changes and retained the same protein content during 13 days of storage in the ovaries of protein-restricted females (*Figure 1E*). Their eventual loss of developmental capacity correlated with the duration of storage (intrinsic aging), but was unaffected by maternal age (*Figure 1F*).



**Figure 1.** Oocytes age reproducibly in a temperature-dependent manner. (**A**) A schematic of the ovarioles that make up a *Drosophila* ovary (above) and the structure of a single ovariole (below) showing the germline stem cells (left) and a string of increasingly mature follicles. Stages 8, 10, and 14 follicles are labeled. Prophase I arrested oocytes undergo meiotic resumption at stage 10, progressing to a secondary arrest point at metaphase I which is maintained until ovulation. (**B**) DAPI stained ovaries from females that were fed 1 day (left), fed 1 day then protein restricted for 1 day (middle), or fed 1 day then protein-restricted 13 days (right). Oocytes are colored as in A, revealing the stable storage of two mature stage 14 oocytes per ovariole. Each mature follicle is about 450 μM in length. (**C**) Eggs laid per day by females containing stored stage 14 follicles, that were provided with males after 3 (green), 6 (blue), 9 (orange) or 12 (red) days. Mating stimulates deposition of the stored oocytes as fertilized embryos. (**D**) Aging curves (days) for follicles stored in vivo at 29°C (magenta), 25°C (green) or 20°C (blue). For each point, stored oocytes were recovered as in (**C**) and the hatch rate determined (N > 100). (**E**) Protein content of mature oocytes that were unstored, stored in vivo for 1 day or for 13 days ('oocyte age'). Protein restriction of mothers does not affect the protein content of stored mature oocytes. (**F**) Hatch rate of embryos developing from fresh mature (stage 14) oocytes from 5-day-old females, mature oocytes stored 14 days (during days 2–16) from 16 day-old females, or fresh mature oocytes from 20-day-old females. Oocyte age during storage, but not maternal age, is associated with reduced hatch rate. Error bars in (**C**), (**E**), and (**F**) denote SD.

## Protein translation declines during oocyte aging

In order to investigate the gene products actively translated by mature oocytes as they aged, we isolated 2, 8, and 12-day-old stage 14 follicles (at 25°C), whose hatch rates are 96%, 89%, and 44%, respectively, removed their follicle cells (see Methods), and performed mRNA-seq and ribosome profiling in triplicate. We added a constant amount of ovarian lysate from *D. pseudoobscura (Dpse)* as a spike-in to allow us to measure changes in translation globally as well as at the single gene level. *Dpse* is sufficiently diverged from *D. melanogaster (Dmel)* that > 97% of 30 nucleotide ribosome

footprints contain at least one polymorphism that can be distinguished by sequencing (see Materials and methods). Ribosome profiling experiments were highly reproducible (*Figure 2—figure supplement 1A–F*). Total mRNA levels, as determined by the ratio of *Dmel* to *Dpse* sequencing reads, did not change significantly between 2 and 12 days of oocyte age (*Figure 2A*). Not only the amount of mRNA, but mRNA composition was also unchanged during aging. RNA-seq transcripts per million (TPM) values from oocytes at day 2 or day 12 of aging correlated very strongly (*Figure 2B*). By contrast, bulk translation in 8 and 12-day-old oocytes was reduced to 59% and 43%, respectively, of levels in 2 day oocytes (*Figure 2C*). Thus, the overall level of translation declines significantly as oocytes age in vivo over a ten-day period, due to widespread declines in translational efficiency (*Figure 2D*).

A reduction in ribosome footprints might theoretically result from an increase in ribosome elongation rates with age rather than a decrease in ribosome initiation. While decreased initiation would lead to decreased overall translation levels, increased elongation would lead to the opposite outcome (increased overall translation). We reasoned that measuring the levels of nascent proteins would allow us to infer relative translation rates and discriminate between these possibilities. We generated *Drosophila* lines expressing the proteasomal substrate R-GFP, which is rapidly degraded by the N-degron pathway (*Dantuma et al., 2000*). Due to its reduced stability, R-GFP staining in the ovary is weaker than the corresponding stable GFP control ('M-GFP') (*Figure 2E*). We then characterized the levels of R-GFP expression in oocytes stored for 1 day or for 12 days, when the level of footprints has declined to ~50% overall of starting levels, with some variation from transcript to transcript. We found that the level of R-GFP fluorescence was reduced by an average of ~30% – as expected if less protein is produced in aged oocytes due to decreased translation initiation (*Figure 2F*). These data strongly support our interpretation that translation levels decline during oocyte aging.

## Aging reduces the translation of all classes of genes

In order to determine whether declining translation in aging oocytes was a general phenomenon or preferentially affected a subset of genes, we analyzed changes to translation and mRNA levels from individual genes. The translation of germline expressed genes such as *Hsp26*, *vtd*, and *me31B*, but not their mRNA levels, reproducibly declined, much like total translation (*Figure 3A*). The changes in translation were not caused by premature egg activation. Genes such as *CycB, CycA, and bora*, whose translation is substantially upregulated at the start of embryogenesis did not increase in aged oocytes (*Figure 3B*). Rather, translational efficiency declined globally (*Figure 2D*, *Supplementary file 1*).

We grouped genes in several functional categories with potential relevance to oocyte aging and compared changes in the mRNA levels and translation levels (*Figure 3C*). Widespread reductions were seen among genes with the GO categories spindle assembly checkpoint, meiotic/mitotic spindle organization, chaperone-dependent protein folding, electron transport chain, and mRNA binding proteins (*Figure 3C*). For each category, the location of several well-known genes is indicated (*Figure 3C*), including genes shown previously to be dose-sensitive for chromosome stability, such as *sub, ncd, nod*, and *SMC1* (*Knowles and Hawley, 1991*; *Moore et al., 1994*; *Subramanian and Bickel, 2008*; *Zhang et al., 1990*).

## A small subset of mRNAs are translationally upregulated in arrested mature oocytes

Mature oocytes stockpile mRNAs, some of which are translated preferentially in oocytes and some of which are translated preferentially during early embryogenesis (*Kronja et al., 2014a*). We reasoned that some genes preferentially translated in arrested oocytes may function to maintain viability during prolonged arrest. In order to identify genes that are preferentially translated in arrested oocytes, we performed ribosome profiling and mRNA-seq experiments on 0–2 hr embryos with *Dpse* ovarian extract spike-in and compared these data to those from arrested oocytes. Total normalized ribosome footprints increased 2.7-fold in developing embryos as compared to mature oocytes stored for two days (*Figure 4A*). While total translation is lower in arrested oocytes than early embryos, we identified a small subset of candidate oocyte 'pilot light genes' (243, representing



**Figure 2.** Stability of mRNA levels and translation in stored stage 14 follicles. (A) Total mRNA per mature follicle normalized to spike-in control after oocyte aging for 2 (blue), 8 (orange) or 12 (purple) days. Differences are not significant (Student's t-test, p=0.78, p=0.48, and p=0.49 for 2 vs. 8, 8 vs. 12, and 2 vs. 12, respectively). (B) Log-Log plot showing high correlation ($R^2$ = 0.97) of mRNA-seq values (TPM, transcripts per million) from stage 14 follicles stored at 25°C for 2 days (96% viability) vs 12 days (44% viability). Equal expression (dashed red line). (C) Total translation levels per mature follicle normalized to spike-in control were compared between oocytes aged for 2

*Figure 2 continued on next page*

*Figure 2 continued*

(blue), 8 (orange) or 12 (purple) days. Differences are significant as shown (Student's t-test). (D) Volcano plot showing global reduction in translation efficiency in 12 day versus 2 day oocytes. (E) R-GFP serves as a reporter of nascent protein levels. Steady state levels of N-end rule proteasomal substrate R-GFP are greatly decreased as compared to stable control M-GFP, consistent with rapid degradation of R-GFP but not M-GFP. Scale bar = 50 μm. (F) Plot showing that R-GFP levels decrease ~30% during oocyte aging, consistent with reduced translation. Error bars in (A) and (C) denote SD.

The online version of this article includes the following figure supplement(s) for figure 2:

**Figure supplement 1.** Reproducibility of ribosome profiling data.

6.0% of oocyte mRNAs) that are preferentially translated during oocyte arrest (*Figure 4B*; *Supplementary file 2*).

The two small heat shock protein chaperones *Hsp26* and *Hsp27* are highly expressed in mature oocytes (*Fredriksson et al., 2012*; *Zimmerman et al., 1983*), and qualified as potential 'pilot light' products since they were translated at higher levels in oocytes than in early embryos (*Figure 4B*). Small heat shock proteins (sHSPs) are also highly expressed during yeast meiosis (*Kurtz et al., 1986*), suggesting a potential conserved function for sHSPs during gametogenesis. In order to test for a role of *Hsp26* and *Hsp27* during prolonged oocyte arrest, we used FRT-mediated recombination to construct a deficiency strain, *Df(sHSP)*, that eliminates *Hsp26* and *Hsp27* (*Figure 4—figure supplement 1A,B*). The strain was backcrossed seven times to the *yw* strain to homogenize the genetic background.

The survival of stored wild type oocytes was compared to stored *Df(sHSP)* homozygous oocytes after between 2 and 14 days of storage in vivo to determine if *Hsp26* and *Hsp27* contribute to oocyte stability. Whereas wild type and *Df(sHSP)/+* oocytes showed normal stability reductions during storage, homozygous Df(sHSP) oocytes lost developmental capacity more quickly (*Figure 4C*). Thus, *Hsp26* and *Hsp27* are important for the survival of oocytes during prolonged storage. We found that *Hsp26* and *Hsp27* are induced in stage 10 egg chambers several hours before the completion of oocyte development (*Figure 4—figure supplement 1B*), consistent with prior data (*Zimmerman et al., 1983*). These data suggest that a subset of genes critical for arrested oocytes are developmentally induced starting just prior to oocyte maturation, in preparation for a prolonged arrest.

Many genes preferentially translated in arrested oocytes are known or are likely to play important roles in the mature oocyte and shortly after the onset of embryogenesis. For example, *Fmr1* is required to optimally maintain stored oocytes (*Greenblatt and Spradling, 2018*). Others including the thioredoxin-like *dhd*, are required to remove sperm protamines following fertilization (*Emelyanov and Fyodorov, 2016*; *Tirmarche et al., 2016*).

In addition, many kinetochore, spindle assembly checkpoint, and meiotic maturation genes including *Ndc80*, *Nek2*, *Zw10*, *gnu*, and *mos* (*Lee, 2003*; *Radford et al., 2015*; *Sagata et al., 1989*; *Uto and Sagata, 2000*; *Williams et al., 1996*) are also preferentially translated in oocytes (*Supplementary file 2*). GO analysis of the 243 genes showed a significantly enrichment for cell cycle-related processes, including mitotic cell cycle, spindle organization, and DNA repair (*Figure 4D,E*). We found that the translation of putative pilot light genes declined during aging slightly more than bulk translation as a whole (*Figure 4—figure supplement 2A,B*).

To investigate whether some functional categories of mRNAs are preferentially stockpiled in advance of oocyte completion, we also gathered information on how gene expression changes as oocyte growth ceases in preparation for storage, ovulation and embryonic development. We carried out RNA-seq and ribosome profiling on the ovaries of young flies 12–16 hr post-eclosion that still lack mature oogenic stages and compared them to day 2 mature oocytes. Genes whose translation is significantly upregulated in mature compared to growing oocytes are summarized in *Supplementary file 3*. These studies were consistent with previous analyses of gene expression during oogenesis and oocyte maturation (*Cui et al., 2013*; *Kronja et al., 2014b*; *Sieber and Spradling, 2015*; *Tootle et al., 2011*). Translation changes late in oogenesis analyzed by gene ontology reflect

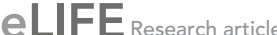

**Figure 3.** Translation is reduced globally during oocyte aging. (**A**) Relative read depths from replicate ribosome footprinting and mRNA-sequencing experiments of *Hsp26, vtd,* and *me31B* from 2-, 8-, and 12-day-old oocytes. Data were normalized to spike-in controls. Relative translational efficiency (right panel) falls 2.5–5 fold between day 2 and day 12. (**B**) Read depths and translational efficiency values are shown as in (**A**) for genes preferentially translated in embryos *CycB, CycA,* and *bora* from 2-day oocytes, 12-day oocytes, and 0–2-hr embryos from non-aged oocytes. (**C**) Heat maps showing reduced translation, but similar mRNA levels, of genes of various GO categories in 12-day oocytes as compared to 2-day oocytes. Gene classes of interest are indicated on the left, along with specific genes on the right; see Flybase for information on each gene (http://flybase.org). Error bars in (**A**) and (**B**) denote SD.



**Figure 4.** Translation of a small group of genes is boosted during oocyte arrest. (A) Normalized total translation levels from ribosome profiling in 0–2 hr embryos compared to oocytes stored for two days. (B) Plot showing that most genes are translated at higher levels in the 0–2 hr embryo than the 2 day stored oocyte. Examples of the 243 more highly translated 'pilot light genes' are labeled in orange. (C) The hatch rate of stored oocytes from wild type (blue), Df(sHSP)/+ (green), or Df(sHSP)/Df(sHSP) (purple) females after indicated storage period ('oocyte age'). Deletion of *Hsp26* and *Hsp27* accelerated the rate of decline during storage (N = 3 at each point). (D–G) GO analysis (PANTHER) of genes with significantly (p<0.01, Student's t-test) increased (D,F) or decreased (E,G) translation in 2- day-old mature oocytes compared to 0–2-hr embryos (D,E) or growing follicles (F,G). Error bars in (A) and (C) denote SD. FDR = false discovery rate.

The online version of this article includes the following figure supplement(s) for figure 4:

**Figure supplement 1.** Deletion of the small heat shock protein gene locus.

**Figure supplement 2.** Genes preferentially translated during oocyte arrest are not protected from widespread age-associated reduced translation efficiency.

completion of follicle growth, reduced ribosomal production, nurse cell dumping, and reactivation of oocyte meiotic progression from diplotene to metaphase I (*Figure 4F,G*).

## Maintaining meiotic spindles limits oocyte longevity

Given the large reductions we observed in the translation of genes related to meiotic spindle organization and the spindle assembly checkpoint, we investigated whether defects in the meiotic spindle could explain the reduced oocyte viability we observed during extended storage. We examined 1-, 7-, and 13-day-old *Drosophila* oocytes expressing α-tubulin-GFP, a construct which has been previously used to analyze meiotic spindles (*Colombié et al., 2008*). We found that the meiotic spindles of day 1 and day 7 oocytes were usually bipolar and highly tapered as previously described (*Theurkauf and Hawley, 1992*) (*Figure 5A and A'*). By day 13 however, many of the spindles were abnormal, such as unipolar, tripolar or fragmented (*Figure 5B–D*). Similar defects are seen in mutants of many of the meiotic spindle maintenance genes (e.g. *sub*, *polo*, and *14-3-3ε*) whose translation declined substantially (>2 fold) during oocyte aging.

In order to determine whether the loss of spindle bi-orientation was the primary cause of oocyte failure, we compared the hatch rate of embryos developing from young (1-day old) or aged (14-day old) oocytes with the proportion of oocytes with bipolar spindles at the same timepoints. We found a striking correlation between the fraction of oocytes able to support development to hatching and the proportion of oocytes with bipolar spindles (*Figure 5E*). Analyzing embryos derived from aged oocytes, we observed errors in chromosomal segregation during the mitotic divisions that follow pronuclear fusion (*Figure 5G,G'*), and meiotic divisions (*Figure 5H,I*). In contrast embryos derived from non-aged oocytes progressed normally through cleavage divisions (*Figure 5F*).

To investigate whether errors of chromosome segregation are the major cause of reduced embryonic viability, we collected embryos derived from unstored (<1 day), 12-day-old and 17-day-old oocytes and analyzed their level of development 4–8 hr after fertilization. Approximately half of embryos derived from oocytes stored for 12 days at 25°C, are able to develop to hatching, but >95% fail to develop after 17 days (*Figure 1D*). 100% of embryos derived from unstored oocytes had progressed to stages 8–12, as expected for normal development (*Figure 5J*). In contrast, the embryos derived from older oocytes showed a bimodal distribution of development. 58% of these embryos from 12-day-old oocytes developed to stages 8–12 like embryos from young oocytes. The other 42% arrested during initial cleavage divisions of pre-blastoderm embryos, failing to progress past the mitotic cell cycles of early embryogenesis preceding zygotic genome activation (*Figure 5J*). In the case of 17-day-old oocytes, almost all derived embryos arrested at pre-blastoderm stages. Only a few percent continued to develop normally (*Figure 5J*). This strong correlation between prolonged storage, lost developmental capacity and embryonic arrest prior to the blastoderm stage further implies that oocyte storage preferentially damages meiotic spindles and the ability to segregate chromosomes accurately to complete meiosis. Consequently, an increasing fraction of oocytes give rise to embryos that undergo chromosome mis-segregation shortly after fertilization leading to lethal aneuploidy.

## Discussion

### *Drosophila* females can be used to study how mature oocytes age during storage in the ovary

We developed a general system for studying the expression and genetic function of genes involved in the aging of completed *Drosophila* oocytes held in the ovary. Using our approach we determined precise aging curves for mature oocytes and showed they varied with temperature. Identifying the genes required for mature oocyte storage in the absence of transcription will elucidate mechanisms that enhance female fertility in many animals, define the limits of these mechanisms, and provide insight into why rare species such as humans are unable to maintain functional oocytes throughout adulthood.

Our studies also address more fundamental questions about the aging of cells that utilize long-lived mRNAs. Oocytes rely heavily on the regulated translation of relatively stable mRNA populations, especially towards the end of egg production. In this they resemble many other cells, including neurons, that utilize relatively stable mRNA at synapses, and male germ cells, which following

**Figure 5.** Stored oocytes lose developmental competence primarily due to problems completing meiosis. Meiotic spindles of oocytes stored for 1, 7 or 13 days at 25˚C were visualized using α-tubulin-GFP (green) and DAPI (magenta) (A–C) or using α-tubulin-GFP alone (A′–C′). Normal bipolar spindles predominate at 1 day (A) but tripolar (B) and unipolar spindles (C) increase, and predominate by 13 days (D) (30 oocytes analyzed per timepoint). (E) Spindle structure correlates closely with oocyte function. (orange bars) The hatch rate of oocytes from wild type animals with the same oocyte and maternal ages measured in parallel. (gray bars) The percentage of oocytes that contain bipolar spindles (measured using α-tubulin-GFP) at the indicated age of storage, produced by females of the indicated ages. (Hatch rates were measured in triplicate and the meiotic spindles of 30 oocytes were analyzed per timepoint). (F–H) Stored oocytes that fail to develop show problems of meiotic completion and preblastoderm arrest. (F) DAPI stained 0–1-hr embryo from an oocyte stored <1 day shows normal cleavage stage nuclei and condensed polar body (arrowhead) visible at the 8 cell stage. (G,G′) 0–1-hr embryo from 12-day-old oocyte shows arrest at the first mitotic division; arrested mitotic spindle (arrow) and polar body (arrowhead). (G′) higher magnification of the spindle in (G) with tubulin-GFP (green) and DAPI (magenta). (H,I) 0–1 hr embryos from a 12-day stored oocyte showing chaotic, arrested meiotic divisions with abnormal, tripolar/fragmented spindles. (J) Stage distribution of embryos from non-stored or 12-day-old or 17 day-old stored oocytes (N>30 at each point). Embryos fell into two categories; embryos from non-aged oocytes developed to stages 8–12 (blue), whereas embryos from 12-day and 17-day oocytes either progressed to stages 8–12 or arrested (orange) during the initial meiotic/mitotic divisions (pre-blastoderm). Scale bars = 10 µM.

meiosis transform into sperm by an elaborate translational program (*Besse and Ephrussi, 2008*; *Fuller, 2016*). Normally, an mRNA turns over in a matter of hours, not days (*Sharova et al., 2009*), and it remains unclear exactly how the functional capacity of mRNAs can be maintained for

extended periods. The close association of long-lasting mRNAs in oocytes, neurons and sperm with P bodies, themselves derived from RNA turnover machinery, and the ability of mRNA to cycle between active and inactive states are likely to play critical roles that can now be studied more easily in a relatively simply and tractable in vivo system, the mature *Drosophila* oocyte.

## A general genomic analysis of translational changes during aging

Our genomic studies reveal the changes in both mRNA and translation levels of essential *Drosophila* genes throughout the aging process. Our data show that despite the potential instability of mRNAs, a measurable decline in mRNA levels is not involved in the loss of oocyte biological function during aging. Using spike-in controls, it was possible to quantitatively compare samples between different time points. We found no significant decline in mRNA levels over the first 12 days of aging at 25˚C.

Despite the preservation of mRNA, there was a pervasive general decrease in mRNA translation that correlated with the loss of oocyte function. Translation must decline either because of changes to mRNAs that reduce their ability to be translated, changes to the ribosomes, RNP granule dysfunction, or alterations in trans-acting factors required for translation. Analyzing the changes in translated proteins during aging did not reveal which of these mechanisms was likely to be responsible. Genes involved in multiple potentially relevant cellular processes undergo significant decreases in translation. These include genes involved in M phase, in meiotic cohesion, and in spindle formation, maintenance and bipolarity. Some of these genes are dose-sensitive (*Knowles and Hawley, 1991*; *Moore et al., 1994*; *Subramanian and Bickel, 2008*; *Zhang et al., 1990*), suggesting that a decrease of two-fold in expression would be enough to generate a phenotype. We observed greatly increased spindle instability as the levels of protein translation fell non-specifically into this range.

## *Drosophila* oocyte decline represents aging in the absence of transcription

An important difference between storage of oocytes within primordial follicles and as full-grown oocytes concerns the status of transcription. Primary oocytes can continue to transcribe genes and repair or replace cellular components as needed, while granulosa cells can divide and replace whole cells if necessary. In contrast, mature oocytes without transcription must rely on translation, which despite the presence of sophisticated RNP-based regulatory machinery undergoes a significant decline in translational efficiency over relatively short periods.

What causes the decline in translation over the course of oocyte aging? One possibility is wear and tear on mRNAs that gradually reduces their ability to undergo translation. Even one cleavage usually inactivates an mRNA and targets it for turnover. Most mRNAs decay within a day or less (*Sharova et al., 2009*) even in growing cells, suggesting that specialized stabilization mechanisms exist during oocyte storage. Unlike proteins, which can be turned over and replaced using mRNA as a template, there is no transcription in mature oocytes and no way to replace damaged mRNA molecules. RNA molecules with expanded trinucleotide repeat sequences can seed the formation of aggregates in a manner analogous to protein aggregation (*Jain and Vale, 2017*; *Querido et al., 2011*). It is unknown if mRNAs are generally susceptible to misfolding and aggregation as has been well-characterized for proteins. We hypothesize that P bodies and stress granules, which form during periods of cellular stress as a response to the accumulation of untranslated mRNAs (*Eulalio et al., 2007*), participate in the long-term preservation of stored mRNAs by decreasing mRNA aggregationor damage.

## Our findings suggest new insight into the strategy of oocyte maintenance in mammals

Our studies have several possible implications for understanding and potentially mitigating the increasing instability of chromosome segregation in mammalian and especially in human oocytes and early embryos. Because of the decades long delay between the onset of meiosis and its completion, some slow decay of an important meiotic process during the primordial follicle stage has been suspected. A logical candidate is the process of sister chromatid cohesion. After forming during premeiotic S phase, meiotic cohesion complexes do not appear to turn over or incorporate freshly synthesized protein subunits (*Revenkova et al., 2010*; *Tachibana-Konwalski et al., 2010*). Reduced levels of the meiotic cohesin complex components Rec8 and Sgo2, which protect cohesin from

separase-mediated cleavage, were observed in aged oocytes (*Chiang et al., 2010*; *Lister et al., 2010*) and interkinetochore distances increase with age (*Merriman et al., 2013*).

However, our results suggest that defects arising during the storage of fully grown oocytes have been under-appreciated as an additional source of meiotic and early embryonic mitotic instability. The production of new transcripts in the oocyte GV strongly drops or ceases after oocytes reach full size, and does not begin again until the 4-cell stage in humans. During this period, oocytes would be largely or entirely dependent on their existing mRNA pool, like stored *Drosophila* mature oocytes (see model, *Figure 6*). Currently, there are insufficient studies using cell marking techniques to follow how long individual full-size mammalian oocytes remain in a quiescent state, what the consequences of late storage are on translation, and whether the average length of mature oocyte storage changes with maternal age and increased incidence of menstrual irregularities. Given the high sensitivity of mature *Drosophila* oocytes to storage shown here, we suggest that a significant fraction of human chromosome instability is caused by the duration of late storage, rather than by defects that occur at the primordial follicle stage. This would imply that the problems of human chromosome instability may be more susceptible to intervention than previously believed.

## Materials and methods

### Oocyte aging assay

Newly eclosed virgin wild type females of indicated genotypes were placed in standard food vials containing added yeast paste made by mixing live yeast with water until the mixture acquires the consistency of peanut butter, but does not trap flies. After feeding on the yeast for 24 hr, flies were transferred to "molasses plates" containing agar-sugar medium (44 g agar, 180 mL molasses, 37 mL

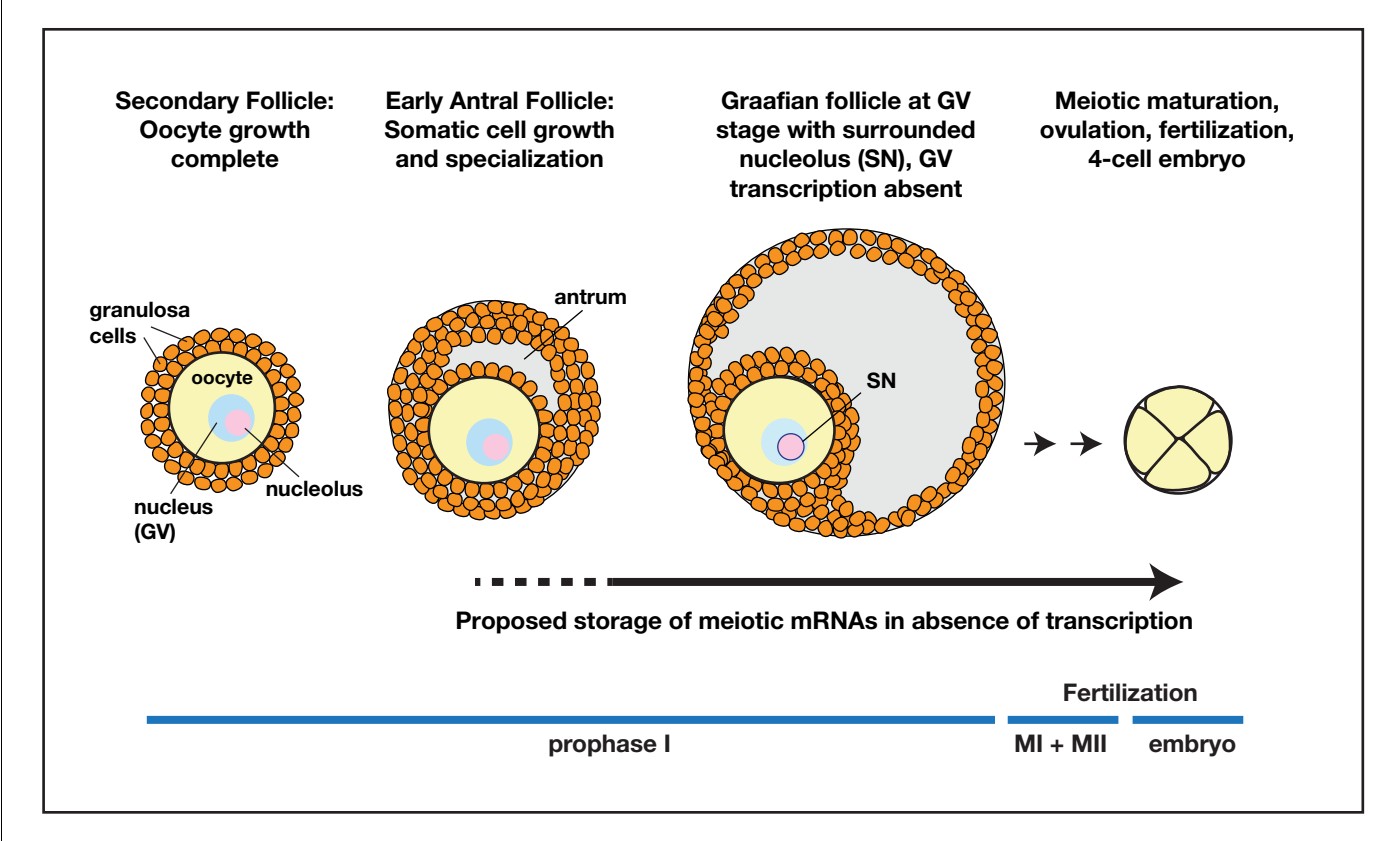

**Figure 6.** Model for prolonged mRNA storage during human oocyte development. Accumulation of meiotic mRNAs occurs prior to the cessation of oocyte growth in pre-antral secondary follicles. Meiotic maturation of prophase I arrested human oocytes requires the translation of mRNAs that have been stored for a prolonged period of development.

in 5% Tegosept, 1112 mL water) to provide humidity, but no edible yeast. After one additional day, ovarioles contain two stage 14 oocytes and are considered to have begun day 1 of quiescence. For study, ovaries were dissected after the desired period of quiescence and the mature stage 14 oocytes were collected. To study oocyte viability, 10 females were transferred to chambers with fresh molasses egg laying plates and 10 males were added. Molasses plates were scored with a needle to increase the number of eggs laid. Males were isolated from females for at least 2 days prior to addition and were aged for 3–8 days from eclosion. Laid embryos were counted and recovered for study after various periods of time. Eggs were collected 16 hr after addition of males and hatch rates were determined 48 hr after collection. Protein measurements were performed using the BCA assay (Pierce).

## Generation of antibodies and deletion alleles of *Hsp26* and *Hsp27*

Antibodies were generated against the C-termini of *Drosophila melanogaster* HSP26 and HSP27 using peptides KLHcarrier-cys-KANESEVKGKENGAPNGKDK and KLHcarrier-cys-APEAGDGKAENG SGEKMETSK respectively (Proteintech) and were used at a concentration of 1:4000. A deletion of the sHSP region was generated via FLP-mediated recombination of FRT-bearing lines d00797 and d05052 from the Harvard Exelixis collection (*Thibault et al., 2004*). Deletion of *Hsp26* and *Hsp27* was confirmed by PCR analysis and immunostaining.

## *Drosophila* ovary and embryo immunostaining

Ovaries were hand-dissected in Grace's Insect Medium (Life Technologies) from flies fed for 3 days with wet yeast paste. Ovaries were fixed in 4% formaldehyde (37% formaldehyde diluted in PBST (0.2% BSA, 0.1% Triton X-100 in 1X PBS) for 12 min. Ovaries were incubated with primary antibodies diluted in PBST with gentle agitation overnight at 4°C. Ovaries were then washed 3 times in PBST for at least 20 min and incubated with secondary antibodies overnight. Ovaries were then washed 3 times with PBST for at least 20 min each, and DAPI (1:20,000-fold dilution of a 5 mg/mL stock) was added to the last wash.

Embryos were dechorionated for 2 min in bleach (50% diluted fresh Clorox bleach). Embryos were fixed for 25 min in a 1:1 mixure of fixative (50 mM EDTA, 9.25% formaldehyde, 1XPBS buffer) and heptane with gentle agitation. The lower fixative layer was removed and an equal volume of methanol was added. Embryos were devitellinized by shaking vigorously by hand for 4 min, removing the heptane layer, and shaking for an addition 1–2 min. Embryos were washed three times in methanol and rehydrated in 50% methanol in PBST, and washed three times in PBST. Embryos were blocked for one hour in PBST and then processed as described for ovaries.

## Ribosome profiling and mRNA-seq library preparation

Ribosome profiling and mRNA sequencing was carried out as described in *Greenblatt and Spradling (2018)* with the following modifications. Aged oocytes were defolliculated by treating ovaries with 5 mg/mL collagenase (Sigma-Aldrich) in PBST for 10 min at room temperature with gentle agitation and then washed three times in PBST. Oocytes were isolated and separated from debris by filtration. Following extraction of defolliculated oocytes or 0–2 embryos with lysis buffer (0.5% Triton X-100, 150 mM NaCl, 5 mM MgCl2, 50 mM Tris, pH 7.5, 1 mM DTT, 20 ug/mL emetine (Sigma-Aldrich), 20 U/mL SUPERaseIn (Ambion), 50 uM GMP-PNP (Sigma-Aldrich)) *Drosophila melanogaster* oocyte extract containing 80 ug RNA was combined with *Drosophia pseudoobscura* whole ovary extract containing 1.6 µg RNA. The combined extract was then processed as in *Greenblatt and Spradling (2018)*.

## Ribosome profiling and mRNA-seq data analysis

Analysis of ribosome profiling and mRNA sequencing data was conducted as in *Greenblatt and Spradling (2018)* with the following modifications. For quantification of bulk mRNA/ribosome footprint levels, adaptor-trimmed reads were mapped to a file containing coding sequences of combined *Dmel* and *Dpse* transcripts using Bowtie v2.3.2 and filtered for only uniquely mapping reads (lines containing string 'NH:i:1') and total reads mapping to either *Dmel* or *Dpse* were counted (*Supplementary file 4*). Unique *Dmel* reads were then re-mapped to the *Dmel* release 6.02 genome with HISAT2 ver2.1.0. Transcripts per million (TPM) values for coding sequences (ribosome profiling)

or exons (mRNA sequencing) were obtained using Stringtie v1.3.5 with the files dmel-CDS-r6.02 gtf or dmel-exons-r6.02.gtf used as a reference annotation for ribosome profiling or mRNA sequencing analysis respectively. Ribosome profiling TPM values from 8 day oocytes, 12 day oocytes, and 0–2 embryo samples were then adjusted by factors of 0.595, and 0.427, and 2.76 respectively to account for bulk changes in translation as determined by the ratios of *Dmel* to *Dpse* reads. Of 26,223 potential 30mer footprints from the top 30 translated *Drosophila* genes in oocytes, we found that 25,324 (97%) sequences contained at least one polymorphism when comparing sequences from orthologous *Dpse* transcripts. Gene ontology analysis was performed using the PANTHER server (*Mi et al., 2019*).

## Acknowledgements

We are grateful to Kamena Kostova, Steve DeLuca, Chenhui Wang, and members of the Spradling lab for support and comments on the manuscript.

## Additional information

### Competing interests

Allan C Spradling: Reviewing editor, *eLife*. The other authors declare that no competing interests exist.

### Funding

| Funder | Author |
| --- | --- |
| Howard Hughes Medical Institute | Allan C Spradling |

The funders had no role in study design, data collection and interpretation, or the decision to submit the work for publication.

### Author contributions

Ethan J Greenblatt, Conceptualization, Data curation, Software, Formal analysis, Validation, Investigation, Visualization, Methodology, Project administration; Rebecca Obniski, Data curation, Formal analysis, Visualization; Claire Mical, Data curation; Allan C Spradling, Conceptualization, Formal analysis, Supervision, Investigation, Methodology

### Author ORCIDs

Ethan J Greenblatt https://orcid.org/0000-0002-3805-0113
Allan C Spradling https://orcid.org/0000-0002-5251-1801

### Decision letter and Author response

Decision letter https://doi.org/10.7554/eLife.49455.sa1
Author response https://doi.org/10.7554/eLife.49455.sa2

## Additional files

### Supplementary files

• Supplementary file 1. Translational and mRNA changes during oocyte aging.

• Supplementary file 2. 'Pilot light' genes translationally upregulated during oocyte arrest.

• Supplementary file 3. Genes upregulated during oocyte maturation.

• Supplementary file 4. Counts of total reads mapping to *Dmel* and *Dpse* for ribosome profiling and mRNA-sequencing experiments.

• Transparent reporting form

## Data availability

Data has been uploaded to BioProjects at NCBI under PRJNA573922.

The following dataset was generated:

| Author(s) | Year | Dataset title | Dataset URL | Database and Identifier |
|---|---|---|---|---|
| Greenblatt EJ, Obniski R, Michael C, Spradling AC | 2019 | Ribosome profiling and mRNA sequencing of aging oocytes in Drosophila. | https://www.ncbi.nlm.nih.gov/sra/PRJNA573922 | NCBI Bioprojects, PRJNA57392 |

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
