## [Decision Letter]

**Acceptance summary:**

The decay of viability of stored oocytes contributes significant to miscarriage and limits reproductive lifespan in humans. This study establishes the *Drosophila* oocyte as a model to study aging of mature oocytes, and provides high-quality data on transcription and translation during the oocytes aging process.

Oocytes are unique cells that can persist for days (or years in the case of humans) in an arrested state. In most animals, oocytes arrest twice during normal development: a first arrest in prophase of meiosis I (which is typically the longer arrest) and a second, typically shorter, arrest after oocyte maturation when the oocyte is in metaphase of Meiosis I or II. How oocytes maintain viability and competency to support embryogenesis during these arrest stages is an area of great interest and practical import.

This paper applies a system the authors previously developed to study *Drosophila* oocytes during the second mature oocyte arrest. This arrest is normally very brief as mature oocytes are normally efficiently ovulated and fertilized. Here, the authors manipulated nutrition and the availability of males to artificially prolong the second arrest by an order of magnitude. They find that, even though transcription has been halted, mRNA levels remain steady in aging oocytes over this period, but that translation rates decrease. They go on to offer evidence that this translational decay plays a role in declining oocyte viability, specifically in the development of abnormal mitotic spindles.

In an effort to identify genes required for viability of arrested oocytes, the authors identify a set of candidate "pilot light" genes that may play a role in maintaining higher level of translation in arrested oocytes compared to embryos. They confirm two of these candidates, *Hsp26* and *Hsp27*, showing that loss of these genes affects oocyte viability, although the precise role of the genes in the process remains to be illuminated.

Overall this study defines a potentially very interesting consequence of aging oocytes after oocyte maturation: a progressive decline in translation rate, and it should be interesting to anyone studying gonadal and germline development as well as infertility. Many questions remain, such as whether this phenotype is specific or reflects a general decline in cell viability, what is the primary cause of the translational slow down and does this process play some role in maintaining viability even as it decreases? This system appears to have great potential for illuminating these and other questions related to the oocyte maturation process.

**Decision letter after peer review:**

Thank you for submitting your article "Prolonged ovarian storage of mature *Drosophila* oocytes dramatically increases meiotic spindle instability" for consideration by *eLife*. Your article has been reviewed by two peer reviewers, and the evaluation has been overseen by Michael Eisen as Reviewing and Senior Editor. The other reviewers have opted to remain anonymous.

I have drafted a review that synthesizes the comments of myself and the two reviewers, who have reviewed and approved it.

The authors have developed a system (previously described in Greenblatt and Spradling, 2018) to study *Drosophila* oocytes during the second arrest ("mature oocyte arrest"). This arrest is normally very brief, as mature oocytes are normally efficiently ovulated and fertilized. By manipulating nutrition and the availability of males, they artificially prolong the second arrest to ~12 days. These oocytes experienced declines in hatch rate with increasing periods of arrest that was correlated with increasing levels of MI spindle abnormalities.

They find that mRNA levels remain steady in aged oocytes over this period, but ribosome footprints gradually decline. They interpret these results as showing that prolonged arrest of mature oocytes leads to a gradual decline in global translation (but see below). The decline is general and includes genes with essential functions such as chaperone dependent protein folding and spindle assembly, consistent with increase in spindle abnormalities as oocytes age.

In an effort to identify genes required for viability of arrested oocytes, the authors identify so-called "pilot light" genes that maintain a higher level of translation in arrested oocytes compared to embryos. They identify two "pilot-light" genes *Hsp26* and *Hsp27* and show that loss of these genes affects oocyte viability, but not spindle assembly. Whether these genes are specifically required in oocytes to maintain viability during arrest, or are more generally required for germ cell viability, is not clear and so the significance of these findings remains uncertain.

Overall this study defines a potentially very interesting consequence of aging oocytes after oocyte maturation: a progressive decline in translation rate. But there are several important issues/questions that need to be addressed:

1) Is the phenomenon specific to aging oocytes? There are several reasons to be concerned that this is not the case.

First the authors utilize starvation to arrest the oocytes. While the premise is that late stage oocytes have already stockpiled the proteins and mRNAs it needs prior to the starvation, previous studies have shown that starvation induces changes that affect the entire organism (for example halting development of earlier oocytes and affecting life span). It is not clear if late stage oocytes are affected by the signaling cascades induced by starvation and if some of the changes observed are the oocytes response to the cue that food will be absent in the environment the oocyte will be deposited. This could be addressed at least partially by exploiting the fact that *Drosophila* females can be made to hold late stage oocytes for 4-5 days without starvation by preventing access to males. Examining such 4-5 day arrested oocytes without starvation would strengthen the argument that translational changes are due specifically to arrest.

Second, the authors point out that in humans oocytes from younger women still showed increased abnormalities when implanted into older woman indicating mother's age may influence developed oocytes as well. Some of the upregulated genes and phenotypes they observed may be influenced by the age of the females (general aging) rather than actual oocyte arrest that would occur in other species. This could be addressed by examining stage 14 oocytes from fed and mated females that are aged 14 days for both translational and cytological assays. Using oocytes from 2 day starved females meant changes may have already occurred in response to starvation and/or arrest, oocytes from non-starved females is the better control.

Finally, it is unclear why the authors were making comparisons to immature follicles and embryos when the focus of the paper is written to focus on changes in arrested oocytes. These comparisons only indicate how a cell in metaphase I is different from cells either early in meiosis or mitosis. Studies have looked at these differences before and were only mentioned in passing. To understand specifically oocyte arrest the authors need to compare oocytes of the same stage but arrested for long periods compared to not arrested stage 14 oocytes. The current studies primarily show what is needed in a metaphase I oocyte.

2) Is the effect specific to translation?

The paper focuses specifically on translation, but it is unclear if the effect is specific to translation. Perhaps the oocytes are just slowly dying and the translational slowdown is a consequence of a general loss in cell viability, drop in ATP levels, etc… Several of the controls suggested in (1) above might address this, but in general it would be very helpful for the authors to demonstrate that the effects on translation are specific and that other metabolic activities are not affected or affected only secondarily, as a consequence of the translational shut down. Otherwise, the data could be interpreted as simply reporting on the slow death of oocytes caused by an artificially extended arrest that may or may not be physiologically relevant.

3) Throughout the paper, the authors rely on ribosomal profiling to measure translation. The data beautifully document that ribosome footprints/mRNA decline with age, but whether this means that translational output also declines is not a foregone conclusion. Ribosome footprints are not necessarily a direct measure of translational output: a decrease in ribosome footprints could mean a decrease in the rate of translation initiation (fewer ribosomes initiating, fewer ribosomes on message, less translational output) *or* an increase in the rate of translation elongation (faster ribosomes, fewer ribosomes on message, more translational output). Without a separate assay to directly measure translational output, it is not possible to distinguish between these two options.

4) There are a number of other prior studies that need to be discussed in detail.

The prior work examining oocyte aging by the Bickel lab needs to be incorporated into the Discussion.

In Fredriksson et al., 2012, *Hsp26* and *Hsp27* were already found to be strongly expressed at 5 and 35 days of age in late stage eggs which the authors need to mention. The authors show expression of these genes by antibody in stage 10 oocytes but the authors needed to show these genes by cytology or quantitative western of stage 14 oocytes that were arrested for 14 days with starvation and from non-starved 14 day females. This would support that these genes are specifically up-translated in oocytes arrested long periods. The past and current data suggest the proteins are loaded prior to stage 14 oocyte arrest.

If the authors are to continue to include the comparisons to mitotic embryos and early stage follicles their results should be compared and contrasted to the results looking at total protein changes by the Orr-Weaver lab.

Specific suggestions:

The authors mention a number of genes that shows changes in translation but fail to always explain what the genes are and/or provide references. Not all the readers interested in oocyte aging would be familiar with *Drosophila* gene names.

Figure 4. To study proteins that are preferentially translated in arrested oocytes and identify "pilot light" genes, the authors compare translation in 2-day old arrested oocytes to early embryos. However, since embryos must have a vastly different translational program than oocytes, it might be more enlightening to compare arrested oocytes to growing oocytes. It seems this experiment has already been done (Figure 4E, 4F) but further analysis of specific genes that are more highly translated in arrested oocytes compared to growing oocytes could potentially give more insight into other "pilot light" genes that contribute to meiotic spindle stability.

Figure 5. In humans, primordial follicles are stored for long periods of time in meiotic prophase before re-entering the cell cycle and briefly pausing again at metaphase II before fertilization. It is my understanding that this metaphase II pause lasts only 1-2 days in vivo. In this study, the authors show that mature *Drosophila* oocytes arrested at metaphase of meiosis I for long periods (> 6 days) exhibit meiotic spindle instability and abnormal chromosome segregation. They imply that this may be relevant to chromosome instability in human females ("Our findings show that storage of highly functional mature oocytes in vivo is sufficient to destabilize chromosome segregation, suggesting that the prolonged storage of mature oocytes may be an important source of meiotic chromosome instability in human females"). Whether mature human oocytes ever experience such a long arrest is not clear. A diagram that clearly compares oogenesis in *Drosophila* versus humans with the different arrest points could help clarify their argument.

Figure 6. The authors hypothesize that "One potential function of sHSPs might be to stabilize the meiotic spindle and thereby extend the functional lifetime of oocytes." However, they then seem to disprove this very statement in Figure 6D. Although loss of *Hsp26* and *Hsp27* has an impact on oocyte maintenance, the mechanism is unknown and as the final figure, this result is a bit disappointing.

---

## [Author Response]

I have drafted a review that synthesizes the comments of myself and the two reviewers, who have reviewed and approved it.The authors have developed a system (previously described in Greenblatt and Spradling, 2018) to study *Drosophila* oocytes during the second arrest ("mature oocyte arrest"). This arrest is normally very brief, as mature oocytes are normally efficiently ovulated and fertilized. By manipulating nutrition and the availability of males, they artificially prolong the second arrest to ~12 days. These oocytes experienced declines in hatch rate with increasing periods of arrest that was correlated with increasing levels of MI spindle abnormalities.They find that mRNA levels remain steady in aged oocytes over this period, but ribosome footprints gradually decline. They interpret these results as showing that prolonged arrest of mature oocytes leads to a gradual decline in global translation (but see below). The decline is general and includes genes with essential functions such as chaperone dependent protein folding and spindle assembly, consistent with increase in spindle abnormalities as oocytes age.In an effort to identify genes required for viability of arrested oocytes, the authors identify so-called "pilot light" genes that maintain a higher level of translation in arrested oocytes compared to embryos. They identify two "pilot-light" genes Hsp26 and Hsp27 and show that loss of these genes affects oocyte viability, but not spindle assembly. Whether these genes are specifically required in oocytes to maintain viability during arrest, or are more generally required for germ cell viability, is not clear and so the significance of these findings remains uncertain.

We thank the reviewers for their comments on our manuscript. We have made changes, including experiments reported in 5 new and 1 modified main figure panels that have addressed all the requested issues as detailed in this guide.

However, we first respond to a major misunderstanding regarding the physiological role of mature egg storage in *Drosophila*. Regarding the in vivo stage 14 arrest we studied the reviewers state:

"This arrest is normally very brief, as mature oocytes are normally efficiently ovulated and fertilized. By manipulating nutrition and the availability of males, they artificially prolong the second arrest to ~12 days.""Otherwise, the data could be interpreted as simply reporting on the slow death of oocytes caused by an artificially extended arrest that may or may not be physiologically relevant."

No evidence supports and much evidence contradicts the idea that mature oocytes in the wild are normally ovulated and fertilized after a "very brief" arrest. We are concerned that a sentence in the Abstract that referred to rapid ovulation by laboratory *Drosophila* maintained in good conditions misled the reviewers regarding the critically important biological function of mature oocyte storage for *Drosophila* in their natural environment.

Natural conditions of *Drosophila* reproduction in the wild differ drastically from the conditions *Drosophila* experience in the laboratory (Markow, 2015, *eLife*), and make a system for storing mature eggs critically important for female reproduction. Adult females in the wild face long odds at reproduction. With adequate protein each female can produce many hundreds of eggs, but in the wild only two progeny enter the next generation on average. Wild females do not have ready access to protein-rich yeast, but have to find this transient, rare resource and compete for access, on multiple occasions. Wild caught females almost never display the large ovaries replete with rapidly developing oocytes such as seen after special feeding in the lab, a sign of chronic protein limitation. In the wild, if stage 14 oocytes are present, the ovarioles usually contain multiple stage 14 oocytes, a sign of the late arrest we studied. Thus, *Drosophila* face the same problem that drove mosquitos to evolve blood feeding.

The ability to store mature oocytes during multi-day searches for ovulation sites probably represents the rate-limiting factor determining female reproductive success. The fact that mature eggs can be stored for 14 days at 25°C (14 times longer than necessary to complete embryonic development) is completely implausible as an unselected "slow death." This is a complex biologically programmed capability. There are extensive preparations during oogenesis for quiescence, including the disassembly of mitochondria, and shut down of oxidative phosphorylation, processes which are then reversed after storage is completed at ovulation. Thus, the ability to store stage 14 oocytes represents an evolved state molded by strong evolutionary pressure to maximize the opportunity for successful reproduction and to avoid wasting resources. Our experiments, which are carried out entirely in vivo and utilize the same protein limitation commonly experienced by wild flies, make it possible to precisely study this critical process of *Drosophila* reproductive biology for the first time. They also serve as a model for understanding oocyte storage in other species, where it likewise often plays a major role.

To ensure that readers understand these fundamental facts that underlie our research program, we have rewritten the Abstract to remove the misleading sentence, and added material in the Introduction describing the critical role of mature oocyte storage in *Drosophila* female reproduction.

Overall this study defines a potentially very interesting consequence of aging oocytes after oocyte maturation: a progressive decline in translation rate. But there are several important issues/questions that need to be addressed:1) Is the phenomenon specific to aging oocytes? There are several reasons to be concerned that this is not the case.First the authors utilize starvation to arrest the oocytes. While the premise is that late stage oocytes have already stockpiled the proteins and mRNAs it needs prior to the starvation, previous studies have shown that starvation induces changes that affect the entire organism (for example halting development of earlier oocytes and affecting life span).

Our protocol does not involve starvation, but protein limitation. Use of the term "starvation" was unfortunate lab jargon that we have now removed. The flies at all times have adequate water, sugar, micronutrients etc. as well as the extensive lipid reserves they stored in their fat bodies during their 24 hr of gorging on pure yeast. It is the lack of available sperm that is inhibiting ovulation, not the physiological condition of the females – they are healthy and begin actively laying immediately following mating even after prolonged protein restriction. While females have enough nutrients to live comfortably under our protocol, they do not have enough protein to make new eggs, and this approach has been used extensively to manipulate oogenesis in past studies without detectably impacting maternal or oocyte physiology. We have clarified these points in the text (subsection “A general method for studying *Drosophila* oocyte aging”).

In order to test whether there are detectable changes in oocytes stored in well-fed vs. protein-restricted animals, we have now compared the protein content of oocytes under each condition. As shown in new Figure 1E, individual oocytes contain 0.27ug of protein under well-fed conditions, and we observe no significant change in protein levels of oocytes stored in animals after 1 or 13 days of protein restriction. These data indicate that there is no signaling leading to reabsorption of protein from mature oocytes after they have formed.

We also looked for potential changes in gene expression induced by our protocol. We compared our ribosome profiling data of stored oocytes in protein restricted animals to prior measurements by the Orr-Weaver lab (Kronja and Orr-Weaver et al., 2014a) of oocytes from well fed animals. As shown in Author response image 1, we found that the translation levels of individual genes from oocytes stored for 2 days in our study are well correlated with translation of genes from non-stored oocytes from the Orr-Weaver study. This is despite significant differences in our ribosome profiling protocol, as oocytes were defolliculated in our experiments prior to lysis while the Orr-Weaver study used oocytes with their follicular layer intact. In addition, ribosome footprints were prepared with MNase in our study rather than RNase I as in the Orr-Weaver study. These data suggest that protein limitation does not substantially alter oocyte gene expression, so that changes seen in older oocytes are an indication of oocyte aging..

Variation in protein availability is a common natural phenomenon for *Drosophila*, and fly's dependence on transient protein sources has caused them to be well adapted to a regime of feast or famine with respect to protein. It is natural to have a burst of oocyte development followed by a period of protein starvation. This type of treatment has been heavily used in past research, for example in studies of ovarian developmental regulation by nutrients (Drummond-Barbosa and Spradling, 2001, and many subsequent publications by the Drummond-Barbosa lab). The slowed developmental rate caused by low protein nutrition causes no differences in the structure or physiology of the oocytes, is reversible, and is mediated by well-studied insulin, Tor and other pathways. Unlike a few insects whose egg size can be affected by maternal nutrition, no such changes occur in *Drosophila melanogaster*.

Finally, it actually does not matter if known or currently unknown pathways are induced as a result of protein limitation. These same effects would be induced in the wild when flies stop laying eggs as a result of environmental protein limitation. Whatever the processes are, they will impact oocyte aging in nature and in our model. We argue that it is critical to first study the effects of oocyte storage on the oocyte, before trying to decipher the pathways that may or may not contribute to those effects.

It is not clear if late stage oocytes are affected by the signaling cascades induced by starvation and if some of the changes observed are the oocytes response to the cue that food will be absent in the environment the oocyte will be deposited. This could be addressed at least partially by exploiting the fact that *Drosophila* females can be made to hold late stage oocytes for 4-5 days without starvation by preventing access to males. Examining such 4-5 day arrested oocytes without starvation would strengthen the argument that translational changes are due specifically to arrest.

We studied whether withholding males alone can cause females to hold eggs sufficiently tightly to allow experimentation for even 4-5 days as claimed. When analyzed quantitatively, we found that withholding males does not create a tight arrest of ovulation. As shown in Author response image 2, in the absence of males but with continual access to a rich protein source, flies lay (and therefore replace) ~15 eggs in 3 days and ~30 eggs by 4 days. Thus, after 3 days 25% of oocytes, and after 4 days, 50% of the initial stage 14 oocytes will have been replaced with younger oocytes, making these time point problematic and all subsequently times entirely unsuitable for study. Since our studies showed that even after 7 days of oocyte storage only about 20% of the MI spindles were defective, it is clear that withholding males is not a viable approach to studying the effects of oocyte storage on spindle function.

**Author response image 2. respfig2:** 

Second, the authors point out that in humans oocytes from younger women still showed increased abnormalities when implanted into older woman indicating mother's age may influence developed oocytes as well. Some of the upregulated genes and phenotypes they observed may be influenced by the age of the females (general aging) rather than actual oocyte arrest that would occur in other species. This could be addressed by examining stage 14 oocytes from fed and mated females that are aged 14 days for both translational and cytological assays. Using oocytes from 2 day starved females meant changes may have already occurred in response to starvation and/or arrest, oocytes from non-starved females is the better control.

We thank the reviewers for raising an interesting question: what is the relative contribution of maternal (general) aging vs. the intrinsic aging of the oocyte. We have now addressed this question in two ways. In order to discriminate between maternal age effects vs. oocyte age effects we compared the hatch rate of newly produced oocytes from young (5 day) versus aged (20 day) females. As shown in new Figure 1F, we found that the hatch rate of newly produced and aged oocytes was not significantly different (98.5% vs. 95.9% respectively, p = 0.114), showing that maternal age alone has no effect on oocyte quality, a reflection of the fact oocytes are continuously produced from stem cells. In contrast, held oocytes aged for 14 days starting when their mother was 2 days old had a drastically reduced hatch rate (13.8%, p = 3.8x10^-4^) even though the female laying them, now 16 days old, was younger than the 20 day old control flies.

In addition, we tested whether maternal age or oocyte storage time influenced the maintenance of bi-oriented meiotic spindles. As shown in new Figure 5E, whereas only 7/30 or 23% of 14 day old held oocytes from 16 day old females have bi-oriented MI spindles, 29/30 or 97% of newly produced MI arrested oocytes from 16 day old females had bi-oriented MI spindles, which is similar to the fraction of bi-oriented MI spindles from newly produced oocytes of young (3 day old) animals (28/30). The% of normal spindles correlated strongly with the hatch rate of equivalently aged vs. newly produced oocytes from the 16 day old females (Figure 5E). We conclude that the reduced ability of long-stored oocytes to support hatching is due to intrinsic oocyte aging leading to the loss of spindle bi-orientation, with little or no contribution from maternal aging. Descriptions of these experiments and their conclusions were added to the text.

Finally, it is unclear why the authors were making comparisons to immature follicles and embryos when the focus of the paper is written to focus on changes in arrested oocytes. These comparisons only indicate how a cell in metaphase I is different from cells either early in meiosis or mitosis. Studies have looked at these differences before and were only mentioned in passing.

We used the immature oocytes in order to compare the proteins translated by a growing oocyte with those produced by a non-growing, mature, "quiescent" oocyte undergoing storage. Oogenesis is a complex developmental program and the meiotic stage differences represent only a part of the program. We wanted to determine if a specific pattern of translation is associated with the ability of oocytes to remain functional while in a non-growing state. Ribosome profiling studies of growing oocytes suitable for comparison were not previously available. These experiments produced a number of candidate genes and the information will be useful to guide future research.

To understand specifically oocyte arrest the authors need to compare oocytes of the same stage but arrested for long periods compared to not arrested stage 14 oocytes. The current studies primarily show what is needed in a metaphase I oocyte.

We agree. The bulk of our paper is a comparison of mature oocytes arrested at stage 14 (metaphase I) that differ only in how long they have been stored in this state within the ovary.

2) Is the effect specific to translation?The paper focuses specifically on translation, but it is unclear if the effect is specific to translation. Perhaps the oocytes are just slowly dying and the translational slowdown is a consequence of a general loss in cell viability, drop in ATP levels, etc… Several of the controls suggested in (1) above might address this, but in general it would be very helpful for the authors to demonstrate that the effects on translation are specific and that other metabolic activities are not affected or affected only secondarily, as a consequence of the translational shut down. Otherwise, the data could be interpreted as simply reporting on the slow death of oocytes caused by an artificially extended arrest that may or may not be physiologically relevant.

Stored oocytes are in a physiologically relevant, natural state as discussed above, so there is no danger that our results are not relevant to the real world. They are replete with enough nutrients to make a first instar larva, so it is hard to see why ATP levels would drop. Females are healthy and lack only the large supply of protein needed to make complete oocytes. Their fat bodies contain a rich supply of lipids for producing ATP and dietary sugars are available as well.

Determining a single “cause” of cellular failure is difficult for any aging system and in fact represents a holy grail of aging research. We went a lot farther than most aging studies by showing that the loss of developmental capacity in our system is due to a specific physiological process, meiotic chromosome segregation, a striking and unexpected result. We also found that translation was reduced broadly and affected all mRNAs to a greater or lesser extent, which raises interesting questions for future studies of aging in the absence of transcription.

It is unlikely that all or even most of these genetic changes had a significant functional impact on oocyte decline in the time frame studied, nor did we make such a claim. Instead, our finding that translation of meiotic spindle genes, several of which are dose sensitive, is reduced over time led to us to determine whether the meiotic spindles of stored oocytes became defective. Determining whether translation changes are solely responsible for the spindle defect or whether other factors are involved is of great interest to us and is an area of active research for a future publication. Our working hypothesis is that reduced translation in a cell that depends entirely on translational control of stored mRNAs will have a substantially negative impact on physiology.

Determining the exact cause of translational decline represents an interesting but entirely separate research subject that is beyond the scope of the current study.

A major conclusion of our paper is that reduced translation of meiotic spindle proteins is correlated with the increased inability of oocytes to maintain MI spindle. We found that with increasing storage time, failure to complete meiosis and/or very early mitotic divisions were the major reasons why oocytes no longer developed into hatching larvae. We have now added additional data to support this claim. New Figure 5E shows that hatch rate is strongly correlated to the fraction of oocytes with bipolar spindles as discussed above, and we have added a timepoint to Figure 5J showing that 98% of embryos from aged 17 day old oocytes fail to develop past the pre-blastoderm stage, consistent with early defects in chromosome segregation leading to aneuploidy. We hypothesize that reduced translation contributes to meiotic spindle defects, both directly due to reduced synthesis of spindle components and spindle assembly checkpoint proteins, as well as indirectly, by affecting cell cycle arrest and re-entry, which are controlled entirely post-transcriptionally. In addition, we provide a rich dataset in which to analyze the basis for translational decline during aging.

The loss of meiotic spindle bipolarity has also been observed both in aged mouse oocytes (Nakagawa and FitzHarris, 2017) and in oocytes from human IVF clinics (McCoy et al., 2015; Haverfield et al., 2017, Human Reproduction). That aged *Drosophila* oocytes and oocytes from aged mammals exhibit similar meiotic spindle polarity defects suggest that our studies are physiologically relevant for aging oocytes in humans and in many other species.

3) Throughout the paper, the authors rely on ribosomal profiling to measure translation. The data beautifully document that ribosome footprints/mRNA decline with age, but whether this means that translational output also declines is not a foregone conclusion. Ribosome footprints are not necessarily a direct measure of translational output: a decrease in ribosome footprints could mean a decrease in the rate of translation initiation (fewer ribosomes initiating, fewer ribosomes on message, less translational output) OR an increase in the rate of translation elongation (faster ribosomes, fewer ribosomes on message, more translational output). Without a separate assay to directly measure translational output, it is not possible to distinguish between these two options.

We have added new experiments supporting our interpretation of the footprint declines. If translation elongation rates increase over time rather than translation initiation decreasing, then protein production will also be increased rather than decreased during aging.

In order to differentiate between decreased initiation vs. increased elongation models, we generated new fly lines which ubiquitously express a rapidly degraded proteasomal N-end rule substrate, R-GFP, which allows one to infer instantaneous translation rates from the levels of nascent protein. We validated these lines by demonstrating that R-GFP levels are substantially lower than a stable M-GFP control line when expressed in *Drosophila* ovaries (new Figure 2E). We found that R-GFP levels were reduced by ~30% in aged (12 day old) vs. young (2 day old oocytes) as shown in new Figure 2F. This is consistent with the overall ~50% reduction in translation which varies to some extent from transcript to transcript, and inconsistent with the model that elongation increases. These data substantially support our (conventional) interpretation of the reduced footprinting results.

4) There are a number of other prior studies that need to be discussed in detail.The prior work examining oocyte aging by the Bickel lab needs to be incorporated into the Discussion.

We mentioned work from the Bickel lab in the Introduction. The Bickel lab prevented access to males as a mechanism of oocyte arrest, but as described above this approach acts only over a few days and is leaky. A major difference between our work and prior work from the Bickel lab is that our study focuses on the aging of stored stage 14 wild type oocytes, while the Bickel lab has studied the effect of aging exclusively in mutant backgrounds. We found that the cause of premature oocyte failure in mutant backgrounds (i.e. sHSP mutants in this paper and *Fmr1* RNAi oocytes in Greenblatt and Spradling, 2018) is often different from the meiotic spindle defects that cause failure of aged stage 14 oocytes in a wild type background. A strong indication of such physiological differences between our work and the research from the Bickel lab despite their focus on genes involved in meiotic segregation is the fact that the age-dependent non-disjunction they observed did not affect the first oocytes laid when repression was relieved, but only affected oocytes fertilized 16-32 hours later (Subramanian and Bickel, 2008). As the authors themselves conclude, this argues that the defects studied were not present in stored mature stage 14 oocytes, but only in much early oocyte stages, which had to mature before becoming detectable. Our studies do not bear on events at these earlier stages.

In Fredriksson et al., 2012, Hsp26 and Hsp27 were already found to be strongly expressed at 5 and 35 days of age in late stage eggs which the authors need to mention. The authors show expression of these genes by antibody in stage 10 oocytes but the authors needed to show these genes by cytology or quantitative western of stage 14 oocytes that were arrested for 14 days with starvation and from non-starved 14 day females. This would support that these genes are specifically up-translated in oocytes arrested long periods. The past and current data suggest the proteins are loaded prior to stage 14 oocyte arrest.

The reviewers are correct to note that sHSP expression in oocytes has been previously noted (both by Fredriksson et al., 2012 and previously by Zimmerman and Meselson et al., 1983). We have added these references to the paper. That small heat shock proteins continue to be translated in arrested oocytes – indeed *Hsp26* is the 11th most highly translated protein in 2 day old arrested oocytes (Supplementary file 1) – is borne out in our ribosome profiling data. In addition, we find that not only are sHSPs expressed but also that *Hsp26/Hsp27* are strongly required in oocytes during prolonged arrest (Figure 4C). Whereas newly produced oocytes lacking sHSPs have hatch rates only slightly lower than wild type, this effect is greatly increased with prolonged storage (99% vs. 86% for 1 day stored oocytes as compared to 90% vs. 26% hatch rate for 10 day stored oocytes). These data support our hypothesis that oocyte “pilot light” genes preferentially translated during arrest as opposed to early embryonic development are indeed important for supporting oocyte viability during prolonged storage.

If the authors are to continue to include the comparisons to mitotic embryos and early stage follicles their results should be compared and contrasted to the results looking at total protein changes by the Orr-Weaver lab.

This paper focuses on the storage of mature oocytes. The ribosome profiling studies of early embryos were simply one effort to find genes that are important for oocyte storage. Because the paper does not study early embryo development it would not be appropriate to enter into an analysis of data from other labs on a peripheral subject. These data are being put into the public domain, allowing such comparisons by any interested researchers studying the oocyte to embryo transition. We are not aware of any major differences, but our experiments provide a much greater depth of coverage and used replicates to improve statistical validity.

Specific suggestions:The authors mention a number of genes that shows changes in translation but fails to always explain what the genes are and/ or provide references. Not all the readers interested in oocyte aging would be familiar with *Drosophila* gene names.

We added a reference to Flybase for information on specific *Drosophila* genes within the categories that we described in the text and called attention to the grouping of these genes in physiological categories in the legend to Figure 3C.

Figure 4. To study proteins that are preferentially translated in arrested oocytes and identify "pilot light" genes, the authors compare translation in 2-day old arrested oocytes to early embryos. However, since embryos must have a vastly different translational program than oocytes, it might be more enlightening to compare arrested oocytes to growing oocytes. It seems this experiment has already been done (Figure 4E, 4F) but further analysis of specific genes that are more highly translated in arrested oocytes compared to growing oocytes could potentially give more insight into other "pilot light" genes that contribute to meiotic spindle stability.

As the reviewer notes, we did compare arrested oocytes to growing oocytes, and to early embryos. The paper does discuss the identification of candidate "pilot light" genes using these datasets. However, this was carried out before we realized that the decline of wild type oocyte viability upon storage was fully explained by declining spindle function. It was before we discovered that the effects of storage on translation are very widespread and affect virtually all transcripts (Figure 2D) We also learned that individual genes are all likely to require functional study. For example, one such difference we observed, high expression of small heat shock proteins during stage 14, was relevant to prolonged storage, but not to MI spindle maintenance. *Hsp26* and *Hsp27* mutants were very different from *Fmr1* mutants. We concluded that further bioinformatic analysis could become tedious in the absence of functional tests and might detract from the major message of the paper regarding spindle instability. We are providing these datasets for mining by other groups and we will continue to analyze genes identified in this manner in the future when genetic analysis can also be provided.

Figure 5. In humans, primordial follicles are stored for long periods of time in meiotic prophase before re-entering the cell cycle and briefly pausing again at metaphase II before fertilization. It is my understanding that this metaphase II pause lasts only 1-2 days in vivo. In this study, the authors show that mature *Drosophila* oocytes arrested at metaphase of meiosis I for long periods (> 6 days) exhibit meiotic spindle instability and abnormal chromosome segregation. They imply that this may be relevant to chromosome instability in human females ("Our findings show that storage of highly functional mature oocytes in vivo is sufficient to destabilize chromosome segregation, suggesting that the prolonged storage of mature oocytes may be an important source of meiotic chromosome instability in human females"). Whether mature human oocytes ever experience such a long arrest is not clear. A diagram that clearly compares oogenesis in *Drosophila* versus humans with the different arrest points could help clarify their argument.

Although widely assumed, the length of the metaphase II pause is not the only relevant issue with regard to late storage. Mammalian oocytes actually reach full size at the end of the primary follicle stage, which is 1-3 weeks before ovulation in mice and 40-85 days before ovulation in humans. During the secondary and especially at the antral stage, they are essentially being stored, while ongoing growth and changes in the follicle appear to be confined to its somatic cells. During much of this time it is likely that oocyte transcription continues. However, during part of the antral stage oocytes develop a SN or "surrounded nucleolus" configuration that is believed to the be an analog of the karyosome, and the nucleus becomes transcriptionally inactive at this point (known as "GV stage oocytes"). We added more information about this aspect of meiotic maturation as well as reference to a recent review (Conti and Franciosi, 2018). This period may be analogous to the stored stage 14 oocytes studied in our manuscript. Unfortunately, it is difficult to measure the length of the GV stage, and it may vary in length between individual follicles. Not enough solid information is currently available to make a reliable timeline to put in our paper. The important conclusion is that in mammals, damage to the spindle leading to meiotic non-disjunction might occur some fraction of the time during late storage, like we see in *Drosophila*.

In addition, humans have a menstrual rather than estrous cycle, dissociating ovulation from sexual behavior. Human MII arrested oocytes are therefore unlikely to be immediately fertilized following ovulation as suggested by the reviewers. Currently it is unknown how long MII arrested oocytes can survive prior to their fertilization. Our data suggests that lengthening the period of time between oocyte maturation and fertilization in which the oocyte must survive in the absence of transcription, which could be caused by delayed fertilization or defective ovulation, may contribute to chromosome instability.

Figure 6. The authors hypothesize that "One potential function of sHSPs might be to stabilize the meiotic spindle and thereby extend the functional lifetime of oocytes." However, they then seem to disprove this very statement in Figure 6D. Although loss of Hsp26 and Hsp27 has an impact on oocyte maintenance, the mechanism is unknown and as the final figure, this result is a bit disappointing.

See above discussion as to difficulty of determining the cause of decline in any aging system. We do not claim to have discovered the cause of aging, but an important previously unappreciated feature of aging that warrants further investigation. We agree and have restructured the paper by moving panels reporting the sHSP data either to Figure 4 or to the supplement.